# Additive-controlled asymmetric iodocyclization enables enantioselective access to both α- and β-nucleosides

Qi Wang[1], Jiayi Mu[1], Jie Zeng[2], Linxi Wan[1], Yangyang Zhong[1], Qiuhong Li[1], Yitong Li[1], Huijing Wang [1]✉ & Fener Chen [1,3,4]✉

β-Nucleosides and their analogs are dominant clinically-used antiviral and antitumor drugs. α-Nucleosides, the anomers of β-nucleosides, exist in nature and have significant potential as drugs or drug carriers. Currently, the most widely used methods for synthesizing β- and α-nucleosides are via N-glycosylation and pentose aminooxazoline, respectively. However, the stereoselectivities of both methods highly depend on the assisting group at the C2′ position. Herein, we report an additive-controlled stereodivergent iodocyclization method for the selective synthesis of α- or β-nucleosides. The stereoselectivity at the anomeric carbon is controlled by the additive (NaI for β-nucleosides; PPh$_3$S for α-nucleosides). A series of β- and α-nucleosides are prepared in high yields (up to 95%) and stereoselectivities (β:α up to 66:1, α:β up to 70:1). Notably, the introduced iodine at the C2′ position of the nucleoside is readily functionalized, leading to multiple structurally diverse nucleoside analogs, including stavudine, an FDA-approved anti-HIV agent, and molnupiravir, an FDA-approved anti-SARS-CoV-2 agent.

Nucleosides play vital roles in enzyme metabolism and regulation, cell signaling, and DNA and RNA synthesis[1]. Structurally, these molecules are composed of nucleobases covalently linked to five-membered sugars (ribose or deoxyribose) through glycosidic bonds at their C1′ positions (anomeric carbons) (Fig. 1a). A nucleoside whose nucleobase at C1′ is *cis* to the hydroxymethyl group at C4′ is referred to as a β-nucleoside, while the analogous *trans*-configured molecule is an α-nucleoside. These configurational differences lead to diverse biological functions and applications. Currently, β-nucleoside analogs are predominantly used to treat viruses and cancers, with almost half of the available antiviral medicines belonging to the β-nucleoside family[2,3]. In addition, 15 FDA-approved β-nucleoside analogs are currently used in clinical practice for anticancer chemotherapy[4,5]. In contrast, α-nucleosides exhibit intriguing physicochemical and biological activities, including high enzyme stabilities, unique parallel double-stranded structures, and inhibitory activities against tumors, bacteria, and plasmodia[6]. Selected β- and α-nucleosides are shown in Fig. 1a.

The coronavirus disease 2019 (COVID-19) pandemic caused by the severe acute respiratory syndrome coronavirus 2 (SARS-CoV-2) is the most devastating global crisis experienced in recent years. To date, the FDA has approved some medicines for the treatment of COVID-19, including remdesivir, molnupiravir, sotrovimab, paxlovid (nirmatrelvir and ritonavir), and others. Two of them, remdesivir (C-nucleoside) and molnupiravir (N-nucleoside), are nucleosidic drugs, while some other nucleoside candidates are being clinically assessed[7–11]. Because nucleoside analogs are crucial components of the arsenal used to battle COVID-19, the development of facile methods for the synthesis of nucleosides with high α/β selectivity is an urgent objective.

N-Glycosylation is the most widely used method for synthesizing nucleosides, especially β-nucleosides (Fig. 1b, blue arrow)[12,13]. The silyl-

[1]Sichuan Research Center for Drug Precision Industrial Technology, West China School of Pharmacy, Sichuan University, Chengdu 610041, China. [2]Pharmaceutical Research Institute, Wuhan Institute of Technology, 430205 Wuhan, China. [3]Engineering Center of Catalysis and synthesis for Chiral Molecules, Department of chemistry, Fudan University, Shanghai 200433, China. [4]Shanghai Engineering Center of Industrial Asymmetric Catalysis for Chiral Drugs, Shanghai 200433, China. ✉e-mail: wanghuijing@scu.edu.cn; rfchen@fudan.edu.cn

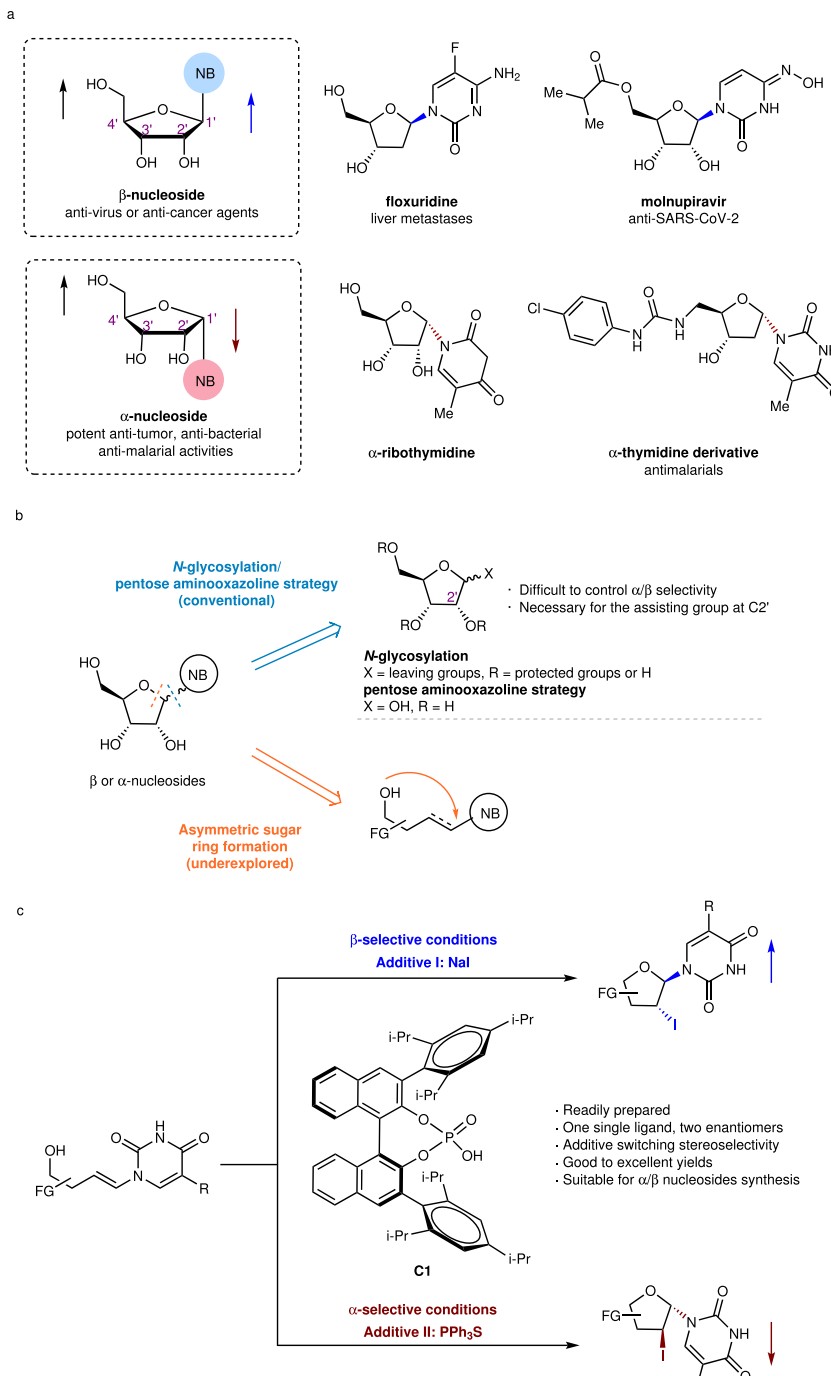

**Fig. 1 | Importance of nucleosides and approaches to synthesize nucleosides. a** Significance of β-nucleosides and α-nucleosides. **b** Strategies to construct nucleosides. **c** Additive-controlled iodocyclization for synthesizing α- and β-nucleosides as two optically pure molecules (this work).

Hilbert-Johnson method developed by Vorbrüggen (Vorbrüggen glycosylation) is the dominant *N*-glycosylation protocol. It uses a strong Lewis acid to catalyze coupling between a per-acetylated sugar synthon and a per-trimethylsilylated nucleobase via a 1', 2'-dioxolenium ion intermediate[14–17]. The α-face of the molecule is blocked to nucleophilic attack, which results in high β-selectivity. Yu et al. coined another representative strategy that involves the gold(I)-catalyzed *N*-glycosylation of a nucleobase with a glycosyl *ortho*-alkynylbenzoate via a glycosyl oxacarbenium intermediate[18]. β-Nucleosides are efficiently synthesized from various alkene- and alkyne-based sugars by Yu glycosylation and its developed methods[18–25]. In most glycosylation, the sugar synthons are fully protected. Hocek et al. reported the

glycosylation of nucleobases with 5'-O-monoprotected ribose or C5'-modified ribose derivatives using modified Mitsunobu conditions to yield β-nucleosides[26,27], whose anomeric selectivity highly depends on the hydroxy group at C2' of the ribosyl donor. Together, the protecting groups of the sugar ring (substrate-dependent), especially the group at the C2' position, determines the stereoselectivity at the anomeric C1' carbon in these methods.

α-Nucleoside synthesis is relatively underexplored compared to its β-nucleoside counterpart. A pentose aminooxazoline usually serves as the key intermediate during the synthesis of an α-nucleoside, as it can be transformed into the pyrimidine α-nucleoside in several steps (Fig. 1b, blue arrow)[6,28–30]. However, poor α/β selectivity

**Table 1 | Reaction condition optimization[a]**

| Entry | Variation from labeled conditions | Yield (%)[b] | ee (%)[c] |
|-------|-----------------------------------|--------------|-----------|
| 1 | none | 89 | 70 |
| 2 | I$_2$ instead of NIS | 58 | 8 |
| 3 | NIP instead of NIS | 84 | 48 |
| 4 | DIDMH instead of NIS | 79 | 45 |
| 5 | no Na$_2$CO$_3$ | 85 | 20 |
| 6 | NaI instead of Na$_2$CO$_3$ | 93 | 96 |
| 7 | PPh$_3$S instead of Na$_2$CO$_3$ | 92 | -51 |
| 8 | PPh$_3$S instead of Na$_2$CO$_3$, CH$_2$Cl$_2$ instead of PhMe:CHCl$_3$ (1:1) | 94 | -90 |

[a]All reactions were performed on 0.1 mmol scale at 40 mM for 8 h. [b]isolated yield. [c]Enantiomeric excess (ee) values were determined by chiral HPLC.

is a limitation of these current methods in the absence of a chimeric participating group at the C2' position of the sugar synthon. The absence of an efficient method for the synthesis of α-nucleosides presents a major roadblock to the further exploration of α-nucleoside bioactivity.

In 2019, Trost et al. reported Pd-catalyzed iodoetherification chemistry for the enantioselective construction of pyrimidine-nucleoside analogs[31]. In particular, this method performed well during the stereoselective synthesis of nucleosides bearing seven-membered sugar rings. This pioneering study suggested that asymmetric sugar-ring formation is an efficient alternative nucleoside synthesis method (Fig. 1b, orange arrow). Asymmetric halocyclization of olefins is an important transformation to construct a wide spectrum of molecular structures[32–38]. However, their applications to the stereoselective synthesis of nucleosides are still underexplored.

Herein, we report a method for the synthesis of α- or β-nucleosides using additive-controlled asymmetric iodocyclization chemistry (Fig. 1c). Stereoselectivity is controlled by the additive (NaI or PPh$_3$S), even while using the same chiral phosphine catalyst. Cyclic products bearing iodine at their C2' positions are readily further functionalized, thereby providing an efficient and atom-economic approach to the synthesis of multifarious nucleosides.

## Results and discussion

Our study commenced with the model combination of achiral alcohol **1a** and N-iodosuccinimide (NIS). After catalysts and solvent screening, we found that chiral phosphoric acid **C1** was the most effective catalyst for this transformation (see Supplementary Tables 6 and 7). Since the desired nucleosidic product **2a** was generated along with a spot corresponding to the spontaneously cyclized product **3a** observed by thin layer chromatography (TLC). To readily determine the yield and stereoselectivity of the reaction, the reaction solution was directly treated with 1,8-diazabicyclo [5.4.0] undec-7-ene (DBU) following iodoetherification, which completely converted the nucleosidic product **2a** into the corresponding cyclic product **3a**. Achiral alcohol **1a** was treated with **C1** (10 mol%), NIS (1.1 eq.), and Na$_2$CO$_3$ (0.1 eq.) in combination of PhMe and CHCl$_3$ (1:1, v·v) at 0 °C, giving the product **R−3a** in 89% yield and 70% ee (Table 1, entry 1). We next, systematically screened the synthetic conditions for halogen sources. However, other halogen sources, such as I$_2$, N-iodophthalimide (NIP), and 1,3-Diiodo-5,5-dimethylhydantoin (DIDMH), provided inferior results (entries 2–4). Further screening revealed that the additive influenced the conversion and enantioselectivity remarkably (see Supplementary Table 9), with NaI identified as the most effective additive for generating **R−3a** (93% yield, 96% ee; entry 6). The absolute configuration of **R−3a** was

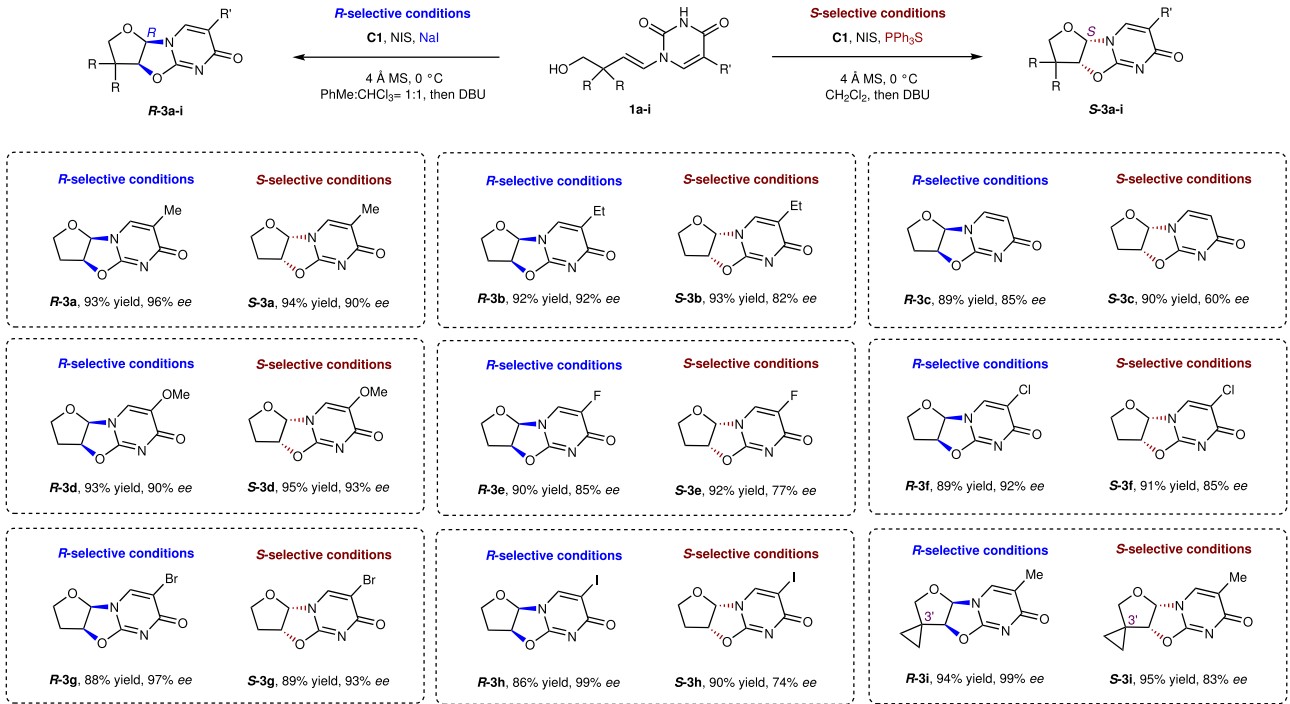

**Fig. 2 | Substrate scope of tetrahydrofuran synthesis.** *R*-selective conditions (blue): **1a-i** (0.1 mmol), **C1** (0.01 mmol), NaI (0.01 mmol), NIS (0.11 mmol), and 4 Å MS (60 mg/mmol), PhMe: CHCl₃ (1.2 mL: 1.2 mL), 0 °C for 8 h, then DBU (0.1 mmol), 0 °C for 30 min; *S*-selective conditions (red): **1a-i** (0.1 mmol), **C1** (0.01 mmol), PPh₃S (0.01 mmol), NIS (0.11 mmol), and 4 Å MS (60 mg/mmol), CH₂Cl₂ (2.4 mL), 0 °C for 8 h, then DBU (0.1 mmol), 0 °C for 30 min.

determined by single-crystal X-ray crystallography[39]. Unexpectedly, the *S*-enantiomer was mainly obtained when PPh₃S was used as the additive (entry 7; 51% *ee*). The reaction was further improved when CH₂Cl₂ was used as the solvent, with *S*−**3a** formed in 90% *ee* and 94% yield (entry 8). We also examined other Lewis bases bearing bulky groups[40], but found that PPh₃S was the most effective *S*-selective additive (see Supplementary Table 9). Though the additive effect has been found in stereoselective glycosylations of six-membered-ring glycosyl donors[41–45], to the best of our knowledge, no additive-controlled stereoselective method to synthesize nucleoside has been reported to date. Herein, we report an additive-controlled asymmetric iodocyclization methodology, with NaI as the *R*-directing (β-directing) additive and PPh₃S as the *S*-directing (α-directing) additive.

With the optimized catalyst **C1** and reaction conditions in hand, we next investigated the substrate scope for the *R*-selective process. In this study, we focused on pyrimidine as the nucleobase for nucleoside construction, since many pyrimidine-nucleoside analogs exhibit impressive bioactivities[46–48], such as efficacy against COVID-19 (molnupiravir), human immunodeficiency virus (HIV, stavudine, zidovudine), herpes simplex virus (HSV, idoxuridine, trifluridine, brivudine), and hepatitis C virus (HCV, sofosbuvir). In addition, it is with worth that pyrimidine α-nucleoside analogs show intuitive bioactivities[49,50]. For instance, α-thymidine analogs inhibit *plasmodium falciparum* thymidylate kinase (*Pf*TMPK), which is promising for the treatment of malaria in the clinic[50].

The use of 5-ethylpyrimidine did not significantly affect the yield or selectivity, giving ***R*−3b** in high yield (92%) and *ee* (92%), as shown in Fig. 2. Thymine ***R*−3c** was also synthesized in excellent yield (89%) but with slightly lower *ee* (85%). Nucleobases bearing other substituents were also examined, with the desired products obtained in high yields (86–94%) and *ee*s (90–99%), apart from ***R*−3e**, which was formed in slightly lower enantioselectivity (85% *ee*). The *S*-selective substrate scope was also investigated, with comparable reactivities and enantioselectivities to those obtained in the *R*-selective processes observed in most cases. However, particular examples, namely ***S*−3c**, ***S*−3e**, and ***S***

−**3h**, were synthesized in high yields and moderate enantioselectivities compared to the desired enantioselectivities of their ***R*−3c**, ***R*−3e**, and ***R*−3h** counterparts.

The introduction of a functional group at the C4 position provides access to nucleoside-like structures that can be further transformed into β- or α-nucleosides. As shown in Scheme 2, we examined a chiral compound bearing a benzyl-ester at the C4 position under the described *R*-/*S*-selective conditions (referred to as β- or α-selective conditions, respectively), which led to **5a** and **6a** in excellent yields (93 and 94%, respectively) with β:α and α:β selectivities of 44:1 and 35:1, respectively. The absolute configurations of **5a** and **6a** were determined by single-crystal X-ray crystallography[51]. In the control reactions, products **5a** and **6a** were obtained in completely racemic forms in the absence of catalysis (neither NaI/**C1** nor PPh₃S/**C1**), which indicates that the chirality of the C4 position does not affect the stereoselectivity of the reaction.

We further investigated the selectivities of processes involving compounds with chiral C4 positions (Fig. 3). The methyl ester substituent was well tolerated, leading to a high yield of **5b** and high β selectivity. In addition, a series of benzyl-ester-substituted alcohols, with various electron-donating or electron-withdrawing substituents on their pyrimidine moieties, exhibited excellent reactivities and stereoselectivities, with the desired β-nucleosides **5c**−**h** obtained in high yields (up to 93%) and selectivities (β:α > 14:1). To closely mimic the nucleoside structure, we introduced a hydroxymethyl group with various protecting groups at the C4 position. All substrates performed well, with the exception of **5l**, which was formed with relatively low stereoselectivity (β:α = 4:1). The phthalimide group was also well tolerated to give **5n** in high yield (90%) and stereoselectivity (β:α = 19:1). The same substrates were also examined under selective conditions, with high yields and selectivities obtained. However, substrates bearing halo-substituents on the pyrimidine were exceptions; **6f**−**h** were obtained in good yields (up to 83%) and moderate stereoselectivities (α:β ratios between 8:1 and 9:1). The absolute configurations of **5l** and its diastereoisomer **6l** were determined by single-crystal X-ray crystallography[52].

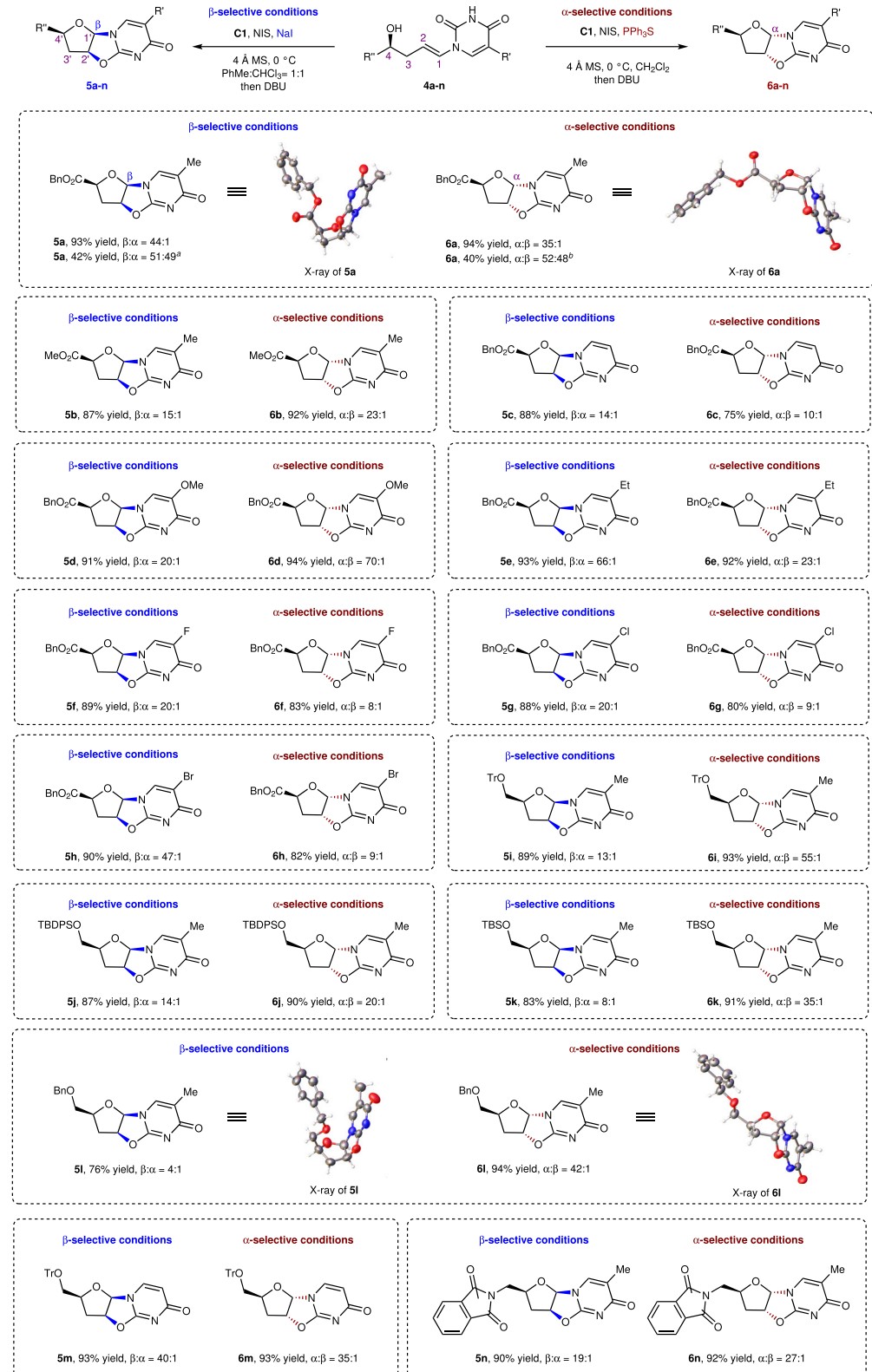

**Fig. 3 | Substrate scope of nucleoside synthesis.** [a]without NaI and **C1**; [b]without PPh₃S and **C1**.

To highlight the scalability of our method for nucleoside synthesis, β-nucleoside analog **5a** and α-nucleoside analog **6a** were synthesized at gram scale from **4a** (Fig. 4). To our delight, yields and selectivities were maintained in these gram-scale reactions. The β-nucleoside analog **5a** was obtained in high yield and β-selectivity (92% yield, β:α > 20:1), while the α-nucleoside analog **6a** was prepared in 93% yield, with α:β > 20:1.

Next, we implemented the derivatizations of the mentioned β- and α-nucleoside analogs to demonstrate their usefulness as synthons (Figs. 5 and 6). As shown in Fig. 5, the iodine in **7** was readily reduced using radical chemistry (Bu₃SnH/AIBN/PhMe) to give tetrahydrofuran derivative **8** in 85% yield. The benzyl carboxylate at the C4′ position in **8** was then reduced by NaBH₄ to provide β-D-ddT **9**, which is an anti-HIV

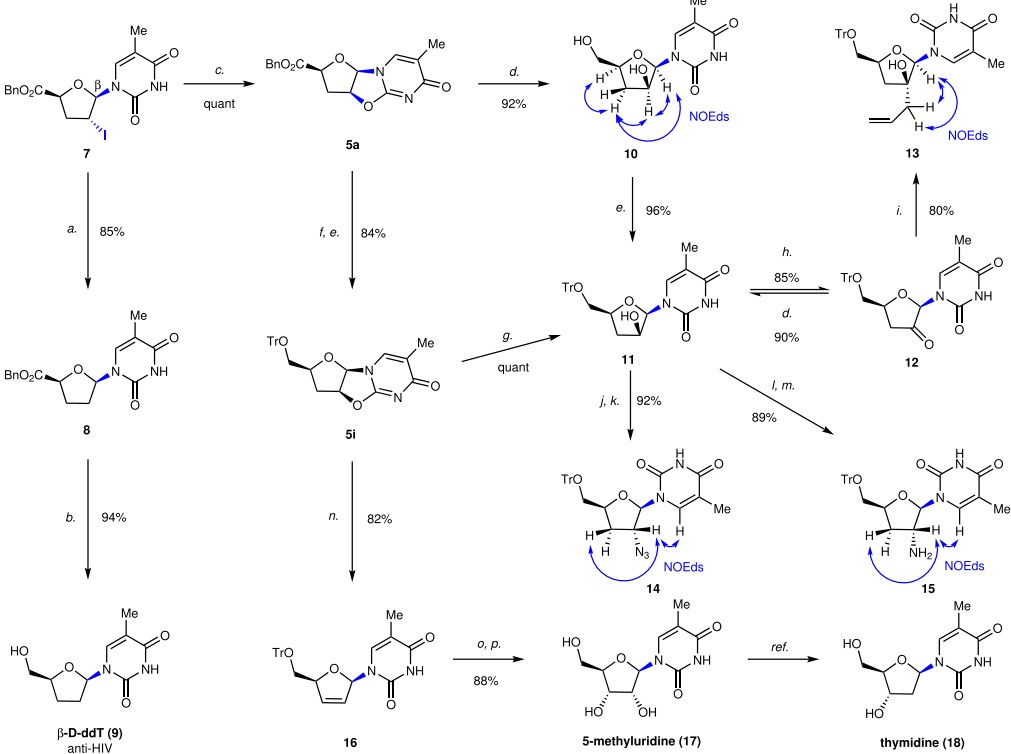

**Fig. 4 | Gram-scale synthesis.** β-Nucleoside analog **5a** and α-nucleoside analog **6a** were synthesized at gram scale from **4a** with high yields and high stereoselectivities.

**Fig. 5 | Derivatizations of β-nucleoside** 7. Conditions: *a*. Bu₃SnH, AIBN, PhMe, reflux, 85%; *b*. NaBH₄, THF:H₂O = 3:1, 0 °C to rt., 94%; *c*. DBU, PhMe:CHCl₃ = 1:1, 0 °C, quant; *d*. NaBH₄, THF:H₂O = 3:1, rt., 92% of **10**, 90% yield of **11** (**12** to **11**); *e*. TrCl, pyridine, DMAP, CH₂Cl₂, reflux; *f*. NaBH₄, THF:H₂O = 3:1, 0 °C, 84% yield over two step (**5a** to **5i**); *g*. NaOH aq. (2.0 M), CH₃CN, rt., quant; *h*. NMO, TPAP, CH₂Cl₂, rt., 85%; *i*. allylmagnesium bromide (1.0 M in THF), THF, −78 °C, 80%; *j*. MsCl, Et₃N, DMAP, CH₂Cl₂, rt.; *k*. NaN₃, DMF, 80 °C, 92% over two steps; *l*. PPh₃, DEAD, phthalimide, THF, rt; *m*. N₂H₄·H₂O, EtOH, 50 °C, 89% over two steps; *n*. *t*-BuOK, DMSO, rt., 82%; *o*. OsO₄, NMO, acetone:H₂O = 4:1, rt.; *p*. 80% AcOH aq., 50 °C, 88% over two steps.

agent candidate[53]. In addition, **5a** was obtained by intramolecular cyclization when **7** was subjected to basic conditions. 3′-Deoxynucleoside analogs have been demonstrated as potential anticancer agents[54]. Thus, we performed the synthesis of 3′-deoxynucleoside **11** from **5a**. NaBH₄-mediated reduction (24 h) of **5a**, in conjunction with ring opening, yielded compound **10** in 92% yield, whose primary alcohol was protected with a triphenylmethyl (Tr) group to generate the 3′-deoxynucleoside analog **11**. When the NaBH₄-mediated reduction of **5a** was carried out in 30 min, only benzyl carboxylic group in **5a** was reduced to yield alcohol intermediate, which was Tr-protected to give **5i** in 84% yield over two steps. Compound **5i** was added into aqueous sodium hydroxide and acetonitrile to give **11**. With the key intermediate **11** in hand, we carried out the synthesis of others 3′-deoxynucleoside analogs **13**, **14** and **15**. Under the conditions of NMO/TPAP, **11** was oxidized to ketone **12** in 85% yield, which could be reversely reduced to **11**. Ketone **12** was then reacted with

allylmagnesium bromide to deliver 3′-deoxynucleoside **13** bearing a quaternary chiral carbon center at the C2′ position. Introduction of the azide group to **11** was accomplished by treatment with MsCl/Et₃N/DMAP followed by NaN₃, leading to 2′-azido-2′,3′-dideoxynucleoside **14** in 92% yield over two steps. Besides, via a two-step transformation, the β-amine group was introduced to **11** at C2′ position, yielding 2′-amino-2′,3′-dideoxynucleoside **15** in 89% yield. In addition, key tricyclic intermediate **5i** could be constructed via the β-selective condition (Fig. 3) to synthesize stavudine, FDA-approved anti-HIV agent (Fig. 7). Ring opening and elimination of **5i** with *t*-BuOK afforded the key intermediate **16**. Stereospecifically, *syn* dihydroxylation of **16** with OsO₄ followed by deprotection afforded 5-methyluridine (**17**), from which thymidine (**18**), a natural nucleoside, can be readily synthesized in three steps, as reported previously[55].

Furthermore, α-nucleoside **19** was also derivatized (Fig. 6); α-D-ddT (**20**) was synthesized from **19** in 77% yield over two steps.

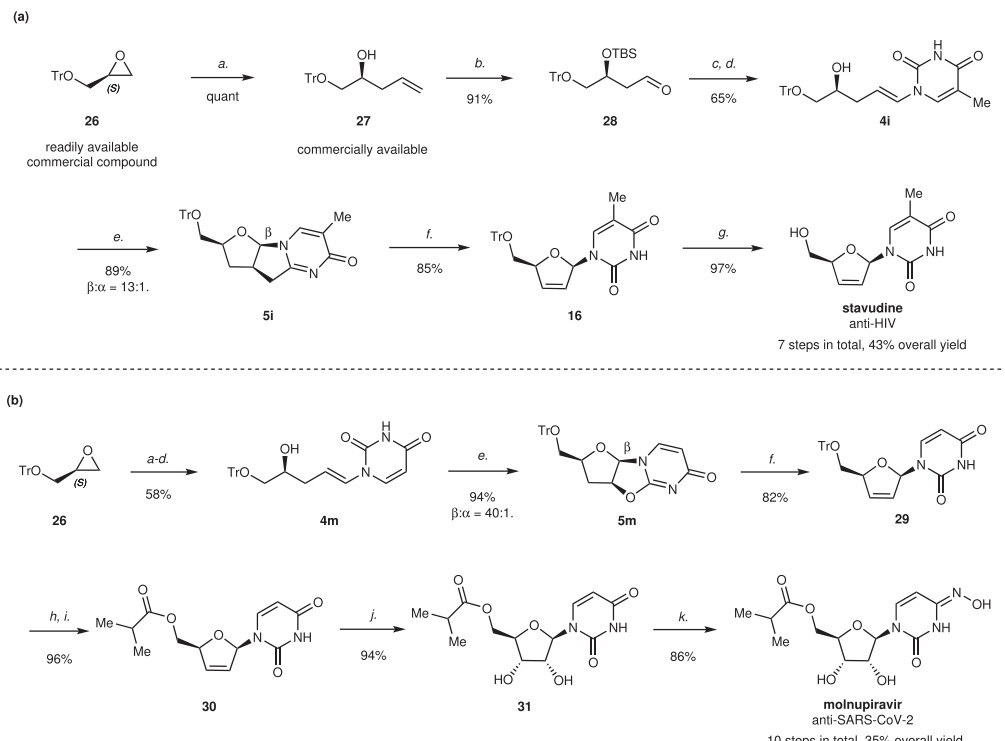

**Fig. 6 | Derivatizations of α-nucleoside 19.** Conditions: a. *a*. Bu₃SnH, AIBN, PhMe, reflux; *b*. NaBH₄, THF:H₂O = 3:1, 0 °C to rt., 77% over two steps; *c*. DBU, CH₂Cl₂, 0 °C, quant; *d*. NaBH₄, THF:H₂O = 3:1, rt., 92%; *e*. NaBH₄, THF:H₂O = 3:1, 0 °C; *f*. TrCl, pyridine, DMAP, CH₂Cl₂, reflux; *g*. *t*-BuOK, DMSO, rt., 60% over three steps; *h*. 80% AcOH aq., 50 °C, 95%; *i*. OsO₄, NMO, acetone:H₂O = 4:1, rt., 62% yield of **24**, 30% yield of **25**.

**Fig. 7 | Synthesis of two FDA-approved agents stavudine and molnupiravir. a** Synthesis of stavudine; **b** Synthesis of molnupiravir. Conditions: *a*. vinylmagnesium bromide (1 M in THF), CuI, THF, −78 °C, 1 h, quant; *b*. TBSCl, imidazole, CH₂Cl₂, 0 °C to rt., then O₃, Et₃N, −78 °C to rt., 91%; *c*. CrCl₂, CHI₃, THF, 0 °C, 82%; *d*. nucleobases (thymine for **4i**, uracil for **4 m**), CuTc, K₃PO₄, DMEDA, DMF, 75 °C, then TBAF, 79% yield of **4i**, 78% yield of **4 m**; *e*. **C1**, NIS, NaI, 4 Å MS, PhMe:CHCl₃ = 1:1, then DBU, 89% of **5i** (β:α = 13:1), 94% of **5 m** (β:α = 40:1); *f*. *t*-BuOK, DMSO, rt., 85% of **16**, 82% of **29**; *g*. 80% AcOH aq., 50 °C, 97%; *h*. 1.0 M HCl aq., MeCN, rt; *i*. isobutyric anhydride, Et₃N, DMAP, MeCN, 96% over two steps; *j*. OsO₄, NMO, acetone:H₂O = 4:1, rt., 94%; *k*. (NH₂OH)₂, H₂SO₄, NH₄HSO₄, imidazole, HMDS, 80 °C, 86%.

Treatment of **19** with DBU afforded **6a**, which was transformed into **21** through slow reduction, while alkene **23** was readily obtained from **6a** in a four-step sequence (reduction, protection of the alcohol, elimination, and deprotection). Moreover, the dihydroxylation of alkene **22** delivered a 2:1 ratio of diastereoisomers **24** and **25**.

To further demonstrate the synthetic usefulness of our method, we applied it to the asymmetric synthesis of two FDA-approved agents, stavudine (anti-HIV) and molnupiravir (anti-SARS-Cov-2). As shown in Fig. 7a, stavudine was readily synthesized from the *S*-trityl glycidyl ether (**26**), a commercially available compound, in 7 steps and in 43% overall yield. Initially, the epoxide of **26** was subjected to ring opening with vinylmagnesium bromide to yield chiral alcohol **27**. Then TBS-protection of alcohol group and oxidative cleavage of terminal alkene group were performed smoothly in one pot, affording aldehyde **28** in 91% yield. Iodination of **28** using $CrCl_2$ and $CHI_3$ gave the *E*-alkenyl iodide as the main product (*E*:*Z* > 10:1), from which the coupling with thymine and removal of the TBS group in one pot occurred to obtain the precursor of iodocyclization **4i** in 65% yield over two steps. Then the β-selective iodocyclization was performed under optimized condition to generate β-nucleoside analog **5i** in 89% yield (β:α = 13:1). Ring opening and elimination of **5i** afforded **16** in the presence of *t*-BuOK in DMSO. After removal of the Tr group, stavudine was obtained. Besides, a ten-step route to synthesize molnupiravir in 35% overall yield was implemented (Fig. 7b). Precursor **4m** was synthesized from **26** in 58% yield in four steps. Then the β-selective iodocyclization of **4m** provided **5m** in 94% yield with excellent β-selectivity (β:α = 40:1). The *t*-BuOK-mediated ring opening and elimination of **5m** in one pot generated **29**, which was coupled with isobutyric anhydride after deprotection, affording **30** in 79% yield over three steps (**5m** to **30**). After subsequent dihydroxylation and hydroamination of **30**, molnupiravir was successfully synthesized.

To gain a thorough understanding of the reaction mechanism, a series of control experiments using substrate **1a** were performed as shown in Supplementary Table 12. To be specific, when the acidic site (-OH) of **C1** was with a methoxy group (Supplementary Table 12, entries 6 and 20), the corresponding product **3a** was nearly racemic (<5% *ee*). Besides, a chiral sodium phosphate **C15** was prepared to catalyze the *R*-selective iodocyclization, which yielded ***R*−3a** in 88% yield with merely 35% *ee* (Supplementary Table 12, entry 7). Thus, the acidic site (-OH) of **C1** is confirmed to play a critical role in the additive-controlled stereoselective iodocyclizations. When the *R*-selective reaction is performed in the absence of NIS (Supplementary Table 12, entry 2), no product was generated. Thus, NIS is determined to be the exclusive iodine source. When the *R*-selective reaction if performed in the absence of NaI (Supplementary Table 12, entry 3), ***R*−3a** was generated an 85% yield with only 20% *ee*. Combined with other NaI dosage screening experiments (Supplementary Table 12, entries 1 and 10–13), it is concluded that the additive NaI cooperates with **C1**, NIS and the substrate to catalyze *R*-selective iodocyclization in a unique manner benefitting stereoselectivity. To elucidate more details, density functional theory (DFT) studies were performed, using alkene **1a** as model substrate (Fig. 8a). The *R*-selective iodocyclization starts favorably from **Int-I** rather than **Int-I'**, which is based on the calculated Gibbs energy difference of 24.1 kcal/mol between **Int-I** and **Int-I'** (**Int-I**: −49.9 kcal/mol, **Int-I'**: −25.8 kcal/mol). It is worthwhile to mention that such intensely exothermic transformations are not observed without the NaI additive (−13.7 kcal/mol, Supplementary Fig. 7), in accordance with the conclusion that NaI is crucial in the catalytic system. Furthermore, we used interaction region indicator (IRI)[56,57] and fuzzy bond order (FBO)[58,59] to analyze the interactions between atoms of **Int-I** (IRI pic. of **Int-I** in Fig. 8a). Interestingly, NaI is found as a centered role that cooperates with **C1**, NIS and substrate **1a** through LP…π interactions and Na-O interactions, providing an excellent stereoselective environment for *R*-selectivity. After electrophilic addition of the iodide anion to the alkene group of **1a**, the **Int-II** is formed. The following

nucleophilic cyclization occurs to generate **P-*R***, with a reaction barrier of 9.6 kcal/mol (**Int-II** to **TS-I**).

In the *S*-selective iodocyclization, Lewis basic $PPh_3S$ was used as an essential additive (Supplementary Table 12, entries 16 and 17). Lewis bases have been reported to activate *N*-halosuccinimides through polarized covalent bonds or through noncovalent interactions[40,60–62]. To understand the interaction type of Lewis basic $PPh_3S$ and NIS in our *S*-selective iodocyclization, we monitored the course of the reaction by $^{31}P$ NMR (Supplementary Fig. 6). It is suggested that $PPh_3S$ activates the electrophilicity of NIS through a weak noncovalent halogen bond, which promotes the forming of the transient halogen-bonded intermediate **PInt-1**. Meanwhile, computational studies of $PPh_3S$-controlled stereoselective iodocyclizations are performed (Fig. 8b). In the first step, $PPh_3S$ enhances the reactivity of NIS via halogen bonding in **Int-III**, which is also supported by $^{31}P$ NMR experiments (Fig. S1). Then the activated **PInt-1** adds to the alkene group of **1a** to form **Int-IV** and **Int-IV'** releasing succinimide (NHS). Because of the large steric hindrance of **C1** with $PPh_3S$, **Int-IV** has lower energy than **Int-IV'** (−12.0 vs −6.2 kcal/mol), facilitating the *S*-selectivity. The O-C bond between **C1** and **1a** is established smoothly to form **Int-V**, which is exothermic by 17.8 kcal/mol (**Int-IV** to **Int-V**) and requires only 1.7 kcal/mol of free energy to activate (**Int-IV** to **TS-II**). In the final step, the cyclization, which is an exothermic transformation (**Int-V** to **P-*S***: −12.9 kcal/mol) via **TS-III**, occurs to produce **P-*S***.

In summary, we developed an additive-controlled stereodivergent iodocyclization method for constructing β- and α-nucleoside analogs with remarkable yields and stereoselectivities that provides an efficient, stereodivergent, and versatile strategy for the synthesis of nucleosides. Meanwhile, pyrimidine-nucleoside products bearing iodine at their C2' positions were shown to be crucial intermediates that can be further functionalized to yield structurally diverse nucleoside analogs. In addition, the anti-HIV drug stavudine and anti-SARS-Cov-2 drug molnupiravir were concisely synthesized using our method. To the best of our knowledge, few existing methods are capable of controlling stereoselectivity merely through using an additive, not to mention nucleoside synthesis. This work not only expands our fundamental chemical understanding. It also contributes to the battle against COVID-19 through facile nucleoside synthesis.

## Methods

### General procedure for *R*-selective (β-selective) reactions

Under an atmosphere of argon, **1a-i** or **4a-n** (0.1 mmol), **C1** (0.01 mmol), NaI (0.01 mmol), and 4 Å MS (60 mg/mmol) were dissolved in anhydrous PhMe: $CHCl_3$ (1.2 mL: 1.2 mL) and stirred at 0 °C for 15 min. After adding *N*-iodosuccinimide (NIS, 0.11 mmol), the reaction mixture was stirred at 0 °C for additional 8 h. After that, DBU (0.1 mmol) was added to the reaction mixture and stirred for 30 min. The solution was diluted with $CHCl_3$ (2 mL) and saturated aqueous ammonium chloride (2 mL). The aqueous layer was extracted with $CHCl_3$ (2 mL × 4). The combined organic layer was washed with brine (2 mL × 4), dried over $Na_2SO_4$ and filtered, concentrated in vacuo. The crude material was purified via silica gel column chromatography to obtain ***R*−3a-i** and **5a-n**.

### General procedure for *S*-selective (α-selective) reactions

Under an atmosphere of argon, **1a-i** or **4a-n** (0.1 mmol), **C1** (0.01 mmol), $PPh_3S$ (0.01 mmol) and 4 Å MS (60 mg/mmol) were dissolved in anhydrous $CH_2Cl_2$ (2.4 mL) and stirred at 0 °C for 15 min. After adding NIS (0.11 mmol), the reaction mixture was stirred at 0 °C for 8 h. After that, DBU (0.1 mmol) was added to the reaction mixture and stirred for 30 min. The solution was diluted with $CHCl_3$ (2 mL) and saturated aqueous ammonium chloride (2 mL). The aqueous layer was extracted with $CHCl_3$ (2 mL × 4). The combined organic layer was

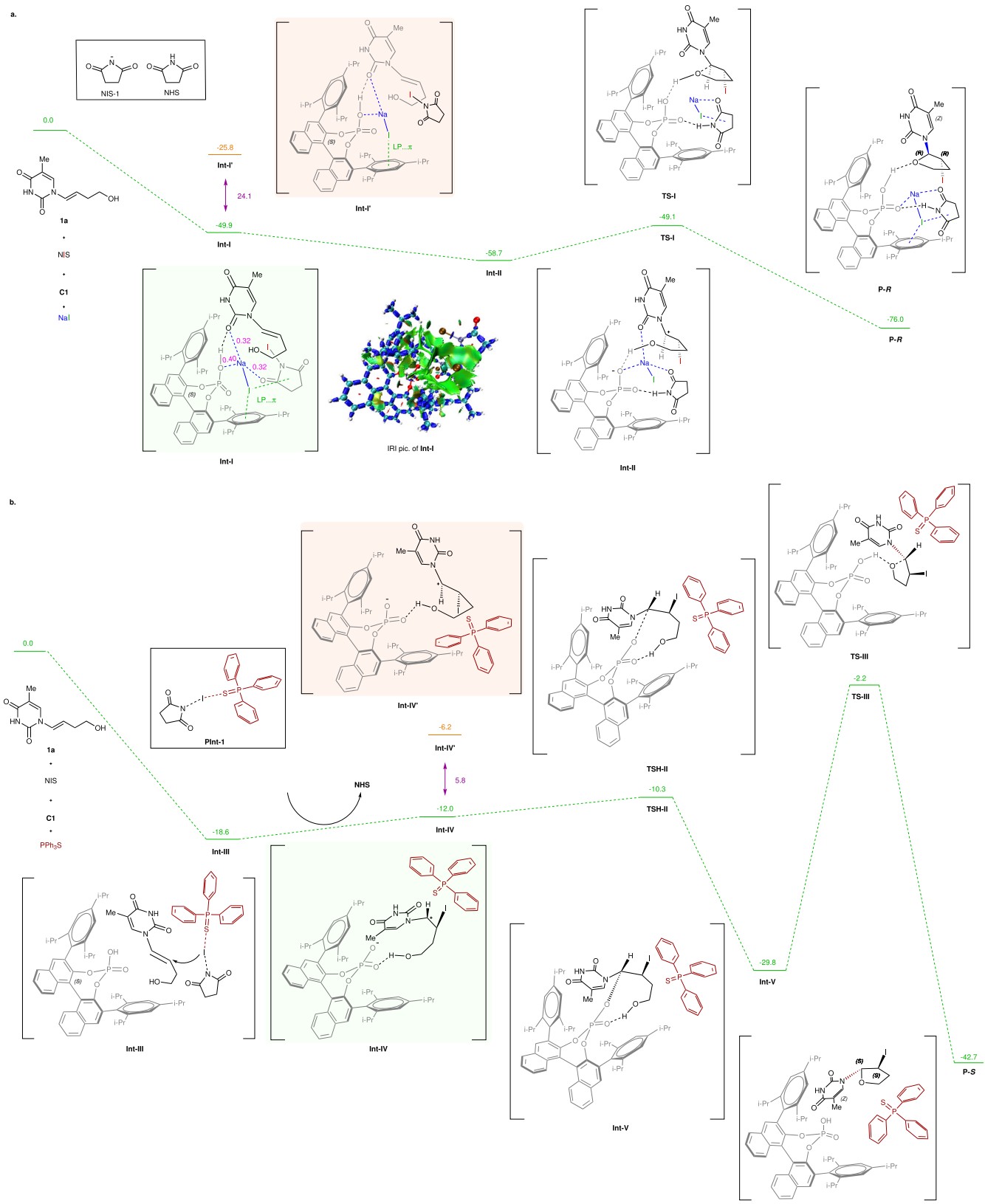

**Fig. 8 | The free-energy profile for the additive-controlled asymmetric iodo-cyclizations. a** Computational studies of NaI-controlled stereoselective iodocy-clizations (*R*-selective/β-selective conditions). **b** Computational studies of PPh₃S-controlled stereoselective iodocyclizations (*S*-selective/α-selective conditions).

washed with brine ($2\,mL \times 4$), dried over $Na_2SO_4$, and filtered, concentrated in vacuo. The crude material was purified via silica gel column chromatography to obtain **S−3a-i** and **6a-n**.

## Experimental data

For the experimental procedures and spectroscopic and physical data of compounds and the crystallographic data of compounds **R−3a**, **5a**, **6a**, **5l**, and **6l**, see Supplementary Methods. For optimization of the reaction conditions and mechanistic studies, see Supplementary Discussion. For NMR spectra of synthetic intermediates, see Supplementary Figs 9–299. For the comparisons of [1]H and [13]C NMR spectra of the known and synthetic β-D-ddT (**9**), stavudine, 5-methyluridine (**17**), and molnupiravir, see Supplementary Tables 15–21. For the HPLC analysis spectra of compounds **R−3a-i**, **S−3a-i**, **5a-n**, and **6a-n**, see Supplementary Figs. 300–322.

## Data availability

The authors declare that the data supporting the findings of this study are available within the paper and its Supplementary Information files, and, also available from the corresponding author upon reasonable request. The Cartesian coordinates are shown in the Supplementary Data 1. Crystallographic data for compound **R−3a** (CCDC No. 2131198), https://www.ccdc.cam.ac.uk/mystructures/structuredetails/164e3c40-0367-ec11-96a5-00505695281c. Crystallographic data for compound **5a** (CCDC No. 2131196), https://www.ccdc.cam.ac.uk/mystructures/structuredetails/c616fb23-0367-ec11-96a9-00505695f620. Crystallographic data for compound **6a** (CCDC No. 2155961), https://www.ccdc.cam.ac.uk/mystructures/structuredetails/bab92800-3b9a-ec11-96aa-00505695281c. Crystallographic data for compound **5l** (CCDC No. 2075890), https://www.ccdc.cam.ac.uk/mystructures/structuredetails/45dec435-0867-ec11-96a9-00505695f620. Crystallographic data for compound **6l** (CCDC No. 2075734), https://www.ccdc.cam.ac.uk/mystructures/structuredetails/1bbd405a-0367-ec11-96a9-00505695f620.

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

## Acknowledgements

We are grateful for the financial support from the National Natural Science Foundation of China (Grant No. 22201192).

## Author contributions

F.C. and H.W. conceived the idea, guided the project. H.W. wrote the manuscript with feedbacks from other authors. Q.W. made the initial observations and analyzed the results. Q.W., J.M., Y.Z., Q.L., and Y.L. explored substrate scope and performed derivatizations. J.Z. performed the density functional theory calculations on the reaction mechanism. L.W. did experiment assistance and analysis for NMR.

## Competing interests
The authors declare no competing interests.
