## [Peer Review File · Nature Communications]

REVIEWER COMMENTS

Reviewer #1 (Remarks to the Author):

The authors report an unprecedented additive-controlled iodocyclization approach for the stereoselective synthesis of α - or β -nucleosides. With NaI as additive, β -nucleosides were obtained in high yields and stereoselectivities, whereas addition of PPh₃S enabled highly stereoselective synthesis of α -nucleosides. By further functionalization of C2' iodine-modified nucleosides, synthesis of a range of diverse nucleoside analogues including stavudine was completed. The manuscript is well prepared, and the experimental procedures and characterization data were given in detail. Although the preparation of the iodocyclization substrates is a bit lengthy, the approach provides a new avenue to the stereocontrolled synthesis of nucleoside analogues. The significance and novelty of this work could arouse broad interest in this field. I would suggest to address the following issues before publication.

The β -nucleosides were actually very convenient to be obtained by participating group effect at the C2' position of the sugar moieties in very few steps compared with the present approach utilizing iodocyclization substrates. Could the authors compare the overall steps and yields of these approaches?

The solvents played a significant effect on the stereoselective synthesis of nucleosides. Could the authors provide explanations on the influence of the solvents on the selectivities of the iodocyclization?

How did the additive NaI function in the NIS-promoted cyclization? Could the authors point out the specific function of the various iodine sources?

Why were the high ee values obtained by NaI instead of KI?

In the hydrogen-bonded transition state, where did the iodine come from, NaI or NIS?

In terms of the synthesis of anti-HIV drug stavudine, the advantages of the approach could be presented compared with traditional method.

The additive-controlled glycosylation have been reported for stereoselective synthesis of various types of sugars, and the recent developments on this aspect should be cited.

Minor points:

Page 4, line 65: "aromatic C1' carbon". I would suggest to replace aromatic with anomeric.

Page 11: "alkyne 20" and "alkyne 21" were not correct. They should be alkenes.

Reference 38: the journal name "PANS" was wrong. Please correct it.

Reviewer #2 (Remarks to the Author):

This manuscript by Wang et al. highlights an asymmetric chiral phosphoric acid-catalyzed approach to both a- and b-nucleoside analogues by modifying the additive in good yields, diastereomeric ratios, and enantiomeric excesses. The manuscript is tidy (with only minor syntactical English errors) and no doubt demonstrates an appreciable body of research; at this time, I recommend this manuscript for acceptance with major revisions.

Nature Communications is a preeminent scientific journal for disseminating results of general interest to the broader scientific community; some manipulations to the text and references are necessary to better convey this, and some additional experiments should be run. I clarify below.

1. A more concise title could be considered. Perhaps “additive-controlled asymmetric iodocyclization enables enantioselective access to both a- and b-nucleosides” would better summarize the results described in this manuscript. “Nucleoside switches” is awkward in wording and to me implies some type of switch, which is not a topic of the work described here.

2. Mention is made to the importance of facilitating access to anti-SARS-CoV-2 drugs or drug carriers (i.e., all of paragraph 2), and the manuscript reports in the last line of the conclusion (the ultimate summary sentence) that this work contributes to the battle against COVID-19; however, no access to anti-SARS-CoV-2 drugs or drug carriers is presented here or a rationale to how this work contributes to the fight against COVID-19 is offered. To better convey this point, access to anti-SARS-CoV-2 drugs should be added to this research. I believe your work could provide elegant access to molnupiravir, for example. Furthermore, a side-by-side overall step and yield count from readily commercially available starting materials using your method to molnupiravir and the current state-of-the-art would be very helpful. This would certainly increase the interest of the manuscript to the general scientific community.

3. A general scheme or at least mention of the step and yield counts to your starting chiral alcohols in the main text (as opposed to somewhat segmented schemes in the SI) would be beneficial to the reader.

4. This work offers scalable access to C3'-deoxy- β -threo-pentonucleofuranosides. Are C3'-deoxy- α -threo-pentonucleofuranosides medicinally relevant? A mention of potential utility would be beneficial. Conversely, a method to access the medicinally relevant C3'-deoxyribofuranonucleosides could be delineated.

5. While your mechanisms are very likely plausible, some references delineating the rationale of the intermediate species you propose (or some basic computational work) would help.

I add a few minor points:

- Stereoselective and protecting group-free access to nucleosides via a 1,2-anhydrosugar has been described and has been used to provide antiviral drugs as well. Pertinent references include *Org Lett* 2015, 18, 46054 and *CEJ*, 2017, 23, 3910 and should be cited.
- Page 2, line 41 – COVID-19 stands for coronavirus disease 2019
- Page 2, line 42 – “Experienced” is a better word than “witnessed” everyone has experienced the pandemic in one capacity or another.
- In the SI, in Scheme S1, you have CHCl₃ above the second arrow; I believe you mean CHI₃.
- The overall quality of English is commendable, only small errors persist. At this time, I believe they are not worth mentioning as the text does require a thorough revision, so the syntactical errors can be addressed in the next iteration.

Reviewer #3 (Remarks to the Author):

This manuscript describes a chiral phosphoric acid catalyzed halo-oxycyclization between ene-urea and NIS. Various substituted 3- tetrahydrofurans can be accessed in high yields with excellent enantioselectivity and diastereoselectivity. Representative products have been transformed into biologically active nucleoside analogues. This paper is written as a methods paper focused on synthetic utility. That’s why very specific 4-aminobut-3-en-1-ol derivatives were selected to undergo cyclization, so the scope of the reaction is quite limited. A stereochemical model is provided, however it is entirely speculative and does not add meaningfully to the paper.

Despite this merit, however, the present study is not impressive in terms of novelty. As mentioned by the authors, this strategy was previously reported by Trost et al (see *J. Am. Chem. Soc.* 2019, 141, 10199–10204) via Pd-catalyzed enantioselective iodoetherification. The authors reported the same enantioselective transformations that can also be carried out using chiral phosphoric acid catalysts. Compared with the reported protocols, the present one is not superior in terms of catalytic efficiency, while an interesting stereodivergent iodocyclization have been described. This precedent work thus reduces the novelty and scientific impact of this contribution.

In addition, this type of reaction is known with bonil phosphoric acid catalysts with enamide derivatives (*ACS Catal.* 2016, 6, 151) and another class of dienes (*Org. Lett.* 2011, 13, 24, 6350, *J. Am. Chem. Soc.* 2017, 139, 4, 1460, *J. Am. Chem. Soc.* 2018, 140, 19, 6039). Therefore, this represents an incremental advance.

Finally, the method was applied to the synthesis of nucleoside analogs, including stavudine, thymidine and β -D-ddT. To demonstrate synthetic utility, it would have been interesting to compare their syntheses with the previous ones (shorter? better overall yield). In addition, the authors used quite hazardous / toxic reagents, such as Bu_3SNH and OsO_4 which can limit industrial applications.

While the reaction is optimized to an outstanding point, the readership of Nature and other top-flight journals expect something more insightful. The catalyst substrate activation method also lacks novel insight as hypothesized. As a result, this work strikes me as rather specialized, and would be a strong communication in Org Lett, Chem Comm, or the equivalent, or potentially JACS as an article (however, more experimental support for the mechanistic hypothesis would be expected). The authors are to be commended for the hard work and persistence that has led to a very nice piece of work. It simply doesn't meet the expected level of impact for a publication in this journal.

Responses to the Reviewers Comments:

Reviewer 1

The authors report an unprecedented additive-controlled iodocyclization approach for the stereoselective synthesis of α - or β -nucleosides. With NaI as additive, β -nucleosides were obtained in high yields and stereoselectivities, whereas addition of PPh₃S enabled highly stereoselective synthesis of α -nucleosides. By further functionalization of C2' iodine-modified nucleosides, synthesis of a range of diverse nucleoside analogues including stavudine was completed. The manuscript is well prepared, and the experimental procedures and characterization data were given in detail. Although the preparation of the iodocyclization substrates is a bit lengthy, the approach provides a new avenue to the stereocontrolled synthesis of nucleoside analogues. The significance and novelty of this work could arouse broad interest in this field. I would suggest to address the following issues before publication.

- The β -nucleosides were actually very convenient to be obtained by participating group effect at the C2' position of the sugar moieties in very few steps compared with the present approach utilizing iodocyclization substrates. Could the authors compare the overall steps and yields of these approaches?

Response 1: *We thank the reviewer for the suggestions. As the reviewer said, the widely applied Vorbrüggen N-glycosylation utilizes the participating group effect of the C2' participating group to achieve β selectivity. It uses a strong Lewis acid to catalyze coupling between a per-acetylated sugar synthon and a per-trimethylsilylated nucleobase via a 1', 2'-dioxolenium ion intermediate. The α -face of the molecule is blocked to nucleophilic attack, which results in high β -selectivity. The protecting groups of sugar ring, especially the group at the C2' position, determines the stereoselectivity at the anomeric C1' carbon in these methods. Therefore, along with other newly developed methods, including Yu glycosylation, it is **substrate dependent**. Otherwise, the method we developed is an **additive-controlled** asymmetric method, which is unprecedented in nucleoside synthesis, to the best of our knowledge, to synthesize α - and β -nucleosides from even the same precursor, with satisfying yields and diastereoselectivities. We believe our method is of great value, especially for the medicinal chemistry.*

Here, we take the synthesis of stavudine, anti-HIV agent, as an example to compare our approach with the reported methods. The reported syntheses of stavudine, commencing with D-ribose, are of 6 to 10 overall steps with 17-60% overall yields (Scheme R1). In our work, we utilized the NaI-additive iodocyclization as the key step to develop an efficient route to synthesize stavudine (7 steps, 43% overall yield), starting from S-trityl glycidyl ether.

We would not claim our method as a game changer, in terms of synthetic efficiency and simplicity. But we would like to humbly point out the versatility of our additive-controlled method and its potential usefulness for medicinal chemistry. Also, the novelty of the unprecedented additive-controlled stereodivergent iodocyclization method for nucleoside synthesis should be emphasized.

Scheme R1. Chemical routes to stavudine synthesis.

- The solvents played a significant effect on the stereoselective synthesis of nucleosides. Could the authors provide explanations on the influence of the solvents on the selectivities of the iodocyclization?

Response 2: We thank the reviewer for the suggestions. The solvents do affect the stereoselectivities in our method (*PhMe* and CHCl_3 beneficial for **R**-selectivity, CH_2Cl_2 beneficial for **S**-selectivity). As suggested in the computational mechanistic study (Figure 2 in main text), in the **R**-selective reactions, the additive NaI plays a centered role that cooperates with **Cl**, **NIS** and substrate **1a** through $\text{LP}\dots\pi$ interactions and Na-O interactions, which require non-polar solvents with lower permittivity, such as *PhMe* and CHCl_3 , to maintain. Whereas, the non-polar CH_2Cl_2 has a relatively higher permittivity, which could be beneficial to activate **NIS** and stabilize the intermediate **Int-IV** composed of **NIS** and PPh_3S .

Solvent	relative permittivity ϵ_r
PhMe	2.38
CHCl_3	4.81
CH_2Cl_2	9.1

- How did the additive NaI function in the NIS-promoted cyclization? Could the authors point out the specific function of the various iodine sources?

Response 3: We thank the reviewer for the question and suggestion. We ran a series of control experiments to explore the mechanism of β -selective iodocyclization (see Figure 2 in main text and Table S14 in SI). When the **R**-selective reaction is performed in the absence of NIS (Table S14, entry 2), no product was generated. Thus, NIS is determined to be the exclusive iodine source. Besides, when the **R**-selective reaction was performed in the absence of NaI (Table S14, entry 3), **R-3a** was generated in 85% yield with only 20% ee. Combined with other NaI dosage screening experiments (Table S14, entries 1 and 10-13), it is concluded that the additive NaI cooperates with **CI**, NIS and the substrate to catalyze **R**-selective iodocyclization, even superior to KI (see Figure S3 in SI), in a unique manner benefitting stereoselectivity.

In addition, the computational studies elucidate the effect of additive NaI in β -selective iodocyclizations. As shown in Figure S2 in SI, the Gibbs energy of reaction to form **Int-I** (-49.9 kcal/mol) is much lower than that of **Int-SI** (-13.7 kcal/mol), indicating the essence of NaI. We further use interaction region indicator (IRI) and fuzzy bond order (FBO) to analyze the interactions between atoms of **Int-I**. As shown in Figure S3 in SI (left), blue represents a covalent bond or ionic bond, and green represents weak interaction. NaI is found as a centered role that cooperates with **CI**, NIS and substrate **1a** through LP... π interactions and Na-O interactions, providing an excellent stereoselective environment for **R**-selectivity.

- Why were the high ee values obtained by NaI instead of KI?

Response 4: We thank the reviewer for the question. As shown in Figure S3 in SI, we analyzed the interaction region indicator (IRI) and fuzzy bond order (FBO) to compare their interactions. Similar to NaI, KI is also found as a centered role that cooperates with **CI**, NIS and substrate through LP... π interactions and K-O interactions. However, K/Na-O interactions in KI is weaker than that of NaI, leading to lower ee values. The fuzzy bond orders (FBO) of K-O bonds in KI are 0.22, 0.24 and 0.26 respectively, which are weaker than those of Na-O bonds in NaI (0.32, 0.32, 0.40 respectively). Moreover, the FBO value of KI (0.39) is smaller than that of NaI (0.51).

- In the hydrogen-bonded transition state, where did the iodine come from, NaI or NIS?

Response 5: We thank the reviewer for the question. We ran a series of control experiments to identify the iodine source, as shown in Table S14 in SI. In the absence of NIS, the **R**-selective iodocyclization failed to occur (Table S14, entry 2), indicating that NIS is the exclusive iodine source. Besides, when the **R**-selective reaction was performed in the absence of NaI (Table S14, entry 3), **R-3a** was generated in 85% yield with only 20% ee. Combined with other NaI dosage screening experiments (Table S14, entries 1 and 10-13), it is concluded that the

additive NaI cooperates with **CI**, NIS and the substrate to catalyze **R**-selective iodocyclization in a unique manner benefitting stereoselectivity.

- In terms of the synthesis of anti-HIV drug stavudine, the advantages of the approach could be presented compared with traditional method.

Response 6: We thank the reviewer for the kind suggestions. The reported syntheses of the stavudine, commencing with D-ribose are of 6 to 10 overall steps (Scheme R2). In our work, we utilized the NaI-additive iodocyclization as the key step to develop an efficient route to synthesize stavudine (7 steps, 43% overall yield), starting from a cheap precursor, S-trityl glycidyl ether. Compared to the reported methods, it seems our synthetic approach does not possess a game-changing advantage. But we want to emphasize the versatility of the additive-controlled method and its potential usefulness for medicinal chemistry. Also, the novelty of the unprecedented additive-controlled stereodivergent iodocyclization method for nucleoside synthesis is noteworthy.

Scheme R2. Chemical routes to stavudine synthesis.

- The additive-controlled glycosylations have been reported for stereoselective synthesis of various types of sugars, and the recent developments on this aspect should be cited.

Response 7: We thank the reviewer for the helpful suggestions. Recent developments in additive-controlled glycosylations have been cited as references 41-45 in page 6 in the revised manuscript.

- Minor points:

Page 4, line 65: “aromatic C1’ carbon”. I would suggest to replace aromatic with anomeric.

Page 11: “alkyne 20” and “alkyne 21” were not correct. They should be alkenes.

Reference 38: the journal name “PANS” was wrong. Please correct it.

Response 8: We thank the reviewer for pointing these out. We have corrected these typographical errors.

Reviewer 2

This manuscript by Wang et al. highlights an asymmetric chiral phosphoric acid-catalyzed approach to both α - and β -nucleoside analogues by modifying the additive in good yields, diastereomeric ratios, and enantiomeric excesses. The manuscript is tidy (with only minor syntactical English errors) and no doubt demonstrates an appreciable body of research; at this time, I recommend this manuscript for acceptance with major revisions.

Nature Communications is a preeminent scientific journal for disseminating results of general interest to the broader scientific community; some manipulations to the text and references are necessary to better convey this, and some additional experiments should be run. I clarify below.

- A more concise title could be considered. Perhaps “additive-controlled asymmetric iodocyclization enables enantioselective access to both α - and β -nucleosides” would better summarize the results described in this manuscript. “Nucleoside switches” is awkward in wording and to me implies some type of switch, which is not a topic of the work described here.

Response 1: We thank the reviewer for the comment and kind suggestion. The title has been changed to ‘Additive-controlled asymmetric iodocyclization enables enantioselective access to both α - and β -nucleosides’ in the revised manuscript.

- Mention is made to the importance of facilitating access to anti-SARS-CoV-2 drugs or drug carriers (i.e., all of paragraph 2), and the manuscript reports in the last line of the conclusion (the ultimate summary sentence) that this work contributes to the battle against COVID-19; however, no access to anti-SARS-CoV-2 drugs or drug carriers is presented here or a rationale to how this work contributes to the fight against COVID-19 is offered. To better convey this point, access to anti-SARS-CoV-2 drugs should be added to this research. I believe your work could provide elegant access to molnupiravir, for example. Furthermore, a side-by-side overall step and yield count from readily commercially available starting materials using your method to molnupiravir and the current state-of-the-art would be very helpful. This would certainly increase the interest of the manuscript to the general scientific community.

Response 2: We thank the reviewer for comments and inspiring suggestion. We synthesized molnupiravir with NaI-controlled iodocyclization as the key step (Scheme 6b in main text). As shown in Scheme R3, the route A and B are semi-synthesis, in which molnupiravir are obtained from uridine and cytidine over 2-5 steps with 17-61% overall yield. Due to the limited resources, uridine or cytidine is synthesized from D-ribose over 3-5 steps. Thus, as to route A and B, the overall synthetic routes are 5-10 steps with <33% overall yield. To avoid use of cytidine or uridine as a feedstock, Merck recently disclosed synthetic routes of molnupiravir taking advantage of ribose as the starting material (route C, 5-10 steps, 30-34% overall yield). Therefore, our synthetic approach of molnupiravir is comparable to the reported routes.

We would like to mention that our additive-controlled iodocyclization is not targeting on synthesis of a particular molecular, but providing a versatile method to readily synthesize a series of nucleosides, including α -nucleosides, β -nucleosides, 2-deoxy-nucleosides, 2,3-dideoxy-nucleosides and so on. We believe this can be an impulse for drug discovery, especially in identifying anti-SARS-CoV-2 drugs, since most of the approved anti-SARS-CoV-2 drugs are nucleosides.

Scheme R3. Chemical routes to molnupiravir synthesis.

- A general scheme or at least mention of the step and yield counts to your starting chiral alcohols in the main text (as opposed to somewhat segmented schemes in the SI) would be beneficial to the reader.

Response 3: We thank the reviewer for the helpful suggestion. Since the synthetic route of the chiral alcohol **4** is relatively simple, description of its synthesis was added in the SI (Page S9). Also, it is showed in Scheme 6 in

main text regarding the synthesis of two FDA-approved agents stavudine and molnupiravir (Page 16 in the revised manuscript), which is hopefully beneficial to the reader.

- This work offers scalable access to C3'-deoxy- β -threo-pentonucleofuranosides. Are C3'-deoxy- α -threo-pentonucleofuranosides medicinally relevant? A mention of potential utility would be beneficial. Conversely, a method to access the medicinally relevant C3'-deoxyribofuranonucleosides could be delineated.

Response 4: We thank the reviewer for the kind comment and suggestion. The C3'-deoxy- α -threo-pentonucleofuranosides' bioactivities, to the best of our knowledge, have not been explored so far. And the C3'-deoxy- α -threo-pentonucleofuranosides have not been synthesized yet. As we mentioned in the introduction part: "The absence of an efficient method for the synthesis of α -nucleosides presents a major roadblock for the further exploration of α -nucleoside bioactivity". Hopefully, the method we report will present a scalable access to the α -nucleosides and facilitate further bioactivity study.

As shown in the Scheme R4, compound **11** and **R1** were subjected to various Mitsunobu reaction conditions to generate the suggested C3'-deoxyribofuranonucleoside, but all failed. Alternatively, introduction of azide group into **11** was accomplished, yielding C2'-azido-C2',C3'-dideoxynucleoside **14** in 92% yield over two steps. Also, C2'-amino-C2',C3'-dideoxynucleoside **15** was obtained in high yield from **11** (also see the Scheme 4 in main text).

Scheme R4. Chemical routes to 3'-deoxypyrimidine nucleosides syntheses.

- While your mechanisms are very likely plausible, some references delineating the rationale of the intermediate species you propose (or some basic computational work) would help.

Response 5: *We thank the reviewer for the kind comment and suggestion. To gain a better understanding of the mechanism of our methodology, a series of control experiments (see Table S14 in SI) and computational studies were performed (Figure 2 in main text), which are all included in the revised paper.*

I add a few minor points:

- Stereoselective and protecting group-free access to nucleosides via a 1,2-anhydrosugar has been described and has been used to provide antiviral drugs as well. Pertinent references include Org Lett 2015, 18, 46054 and CEJ, 2017, 23, 3910 and should be cited.

Response 6: *We thank the reviewer for the helpful suggestions. In accordance with the reviewer's suggestion, stereoselective and protecting group-free access to β -nucleosides via 1,2-anhydrosugar has been well developed. In the revised manuscript, we stated 'In most glycosylation, the sugar synthons are fully protected. Hocek et al. reported the glycosylation of nucleobases with 5'-O-monoprotected ribose or C5'-modified ribose derivatives using modified Mitsunobu conditions to yield β -nucleosides, whose anomeric selectivity highly depends on the hydroxy group at C2' of the ribosyl donor' in page 4. Also, the suggested papers were cited (citation 26 and 27) in the revised manuscript.*

- Page 2, line 41 – COVID-19 stands for coronavirus disease 2019

- Page 2, line 42 – “Experienced” is a better word than “witnessed” everyone has experienced the pandemic in one capacity or another.

Response 7: *We thank the reviewer for the helpful comments and kind suggestions. These points have been modified.*

- In the SI, in Scheme S1, you have CHCl_3 above the second arrow; I believe you mean CHI_3 .

Response 8: *We thank the reviewer for pointing this out. We have corrected this typographical error.*

- The overall quality of English is commendable, only small errors persist. At this time, I believe they are not worth mentioning as the text does require a thorough revision, so the syntactical errors can be addressed in the next iteration.

Response 9: *We thank the reviewer for the kind comment and suggestion. In the modified manuscript, further proofreading has been done, with the modifications highlighted yellow.*

Reviewer 3

- This manuscript describes a chiral phosphoric acid catalyzed halo-oxycyclization between ene-urea and NIS. Various substituted 3-tetrahydrofurans can be accessed in high yields with excellent enantioselectivity and diastereoselectivity. Representative products have been transformed into biologically active nucleoside analogues. This paper is written as a methods paper focused on synthetic utility. That's why very specific 4-aminobut-3-en-1-ol derivatives were selected to undergo cyclization, so the scope of the reaction is quite limited.

Response 1: *We thank the reviewer for the extremely helpful comments. In the current method we report, which utilized stereodivergent iodocyclization to synthesize ribofuranonucleosides, 4-aminobut-3-en-1-ol derivatives were adopted as the precursors. As the reviewer stated, 'various substituted 3-tetrahydrofurans can be accessed in high yields with excellent enantioselectivity and diastereoselectivity' even in a stereodivergent manner. With this paper, we are very excited to report this finding. In addition, the following research underwent in our lab has proved the azanucleosides could also be obtained via similar cyclization strategy.*

- A stereochemical model is provided, however it is entirely speculative and does not add meaningfully to the paper.

Response 2: *We thank the reviewer for the comment. To gain a better understanding of the mechanism of our methodology, a series of control experiments (see Table S14 in SI) and computational studies were performed (Figure 2 in main text), which are all included in the revised paper.*

-Despite this merit, however, the present study is not impressive in terms of novelty. As mentioned by the authors, this strategy was previously reported by Trost et al (see J. Am. Chem. Soc. 2019, 141, 10199–10204) via Pd-catalyzed enantioselective iodoetherification. The authors reported the same enantioselective transformations that can also be carried out using chiral phosphoric acid catalysts. Compared with the reported protocols, the present one is not superior in terms of catalytic efficiency, while an interesting stereodivergent iodocyclization have been described. This precedent work thus reduces the novelty and scientific impact of this contribution. In addition, this type of reaction is known with bonil phosphoric acid catalysts with enamide derivatives (ACS Catal. 2016, 6, 151) and another class of dienes (Org. Lett. 2011, 13, 24, 6350, J. Am. Chem. Soc. 2017, 139, 4, 1460, J. Am. Chem. Soc. 2018, 140, 19, 6039). Therefore, this represents an incremental advance.

Response 3: *We thank the reviewer for the helpful comments. In the previously submitted manuscript, Prof. Trost's excellent and pioneering work (J. Am. Chem. Soc. 2019, 141, 10199–10204), which inspired our work, was remarkably cited and described. Humbly, we would like to point out the highlights in our work:*

1) *Different catalysts mean dissimilar catalytic systems. Prof. Trost's work utilized Cp(allyl)Pd as the catalyst. While in our work, the catalysts are composed of chiral phosphoric acid catalyst with NaI or PPh₃S, which*

achieves stereodivergency in surprise. Two different additives (NaI and PPh₃S) led to different anomeric selectivity. To our best knowledge, such additive-controlled stereodivergent method has not been reported in asymmetric halocyclization of olefins so far.

- 2) The revised paper includes a series of control experiments (see Table S14 in SI) and computational studies recently performed (Figure 2 in main text), which elucidate the mechanism of our method. It was revealed that NaI cooperates with **CI**, NIS and the substrate to catalyze **R**-selective iodocyclization in a unique manner, which is even superior to KI (Figure S3 in SI). The catalytic effect of NaI in our method is, in our opinion, quite novel and illuminating for further study.
- 3) Prof. Trost's work basically focused on the synthesis of nucleoside analogs bearing seven-membered sugar rings. Our method can be applied to the synthesis of ribofuranonucleosides with biological effect, even in an additive-controlled stereodivergent manner. We believe this is of great interest to the medicinal chemistry.

- Finally, the method was applied to the synthesis of nucleoside analogs, including stavudine, thymidine and β -D-ddT. To demonstrate synthetic utility, it would have been interested to compare their syntheses with the previous ones (shorter? better overall yield).

Response 4: We thank the reviewer for the extremely helpful comments and kind suggestions. Here, we take the synthesis of stavudine, anti-HIV agent, as an example to compare our approach with reported methods. The reported syntheses of the stavudine, commencing with D-ribose, are of 6 to 11 overall steps with 17-60% overall yields (Scheme R5). In our work, we utilized the NaI-additive iodocyclization as the key step to develop an efficient route to synthesize stavudine (7 steps, 43% overall yield), starting from S-trityl glycidyl ether.

We would not claim our method as a game changer, in terms of synthetic efficiency and simplicity. But we would like to humbly point out the versatility of our additive-controlled method and its potential usefulness for medicinal chemistry. Also, the novelty of the unprecedented additive-controlled stereodivergent iodocyclization method for nucleoside synthesis is noteworthy.

Scheme R5. Chemical routes to stavudine synthesis.

- In addition, the authors used quite hazardous / toxic reagents. such as Bu_3SnH and OsO_4 which can limit industrial applications.

Response 5: We thank the reviewer for the helpful comments. As the reviewer stated, Bu_3SnH and OsO_4 are unacceptable in industrial production. Meanwhile, we are constantly working on solving these issues using green chemistry.

- While the reaction is optimized to an outstanding point, the readership of Nature and other top-flight journals expect something more insightful. The catalyst substrate activation method also lacks novel insight as hypothesized. As a result, this work strikes me as rather specialized, and would be a strong communication in Org Lett, Chem Comm, or the equivalent, or potentially JACS as an article (however, more experimental support for the mechanistic hypothesis would be expected). The authors are to be commended for the hard work and persistence that has led to a very nice piece of work. It simply doesn't meet the expected level of impact for a publication in this journal.

Response 6: We thank the reviewer for the kind comments. We ran a series of control experiments (see Table S14 in SI) and carried out computational studies to demonstrate the reaction mechanism (Figure 2 in main text), which is hopefully helpful to improve the scholar presentation and impact of our work.

REVIEWERS' COMMENTS

Reviewer #1 (Remarks to the Author):

The authors have addressed the concerns of the reviewers. I recommend acceptance of the manuscript for publication.

Reviewer #2 (Remarks to the Author):

I thank the authors for their corrections and additions; at this time, I recommend the article for publication in Nature Communications. This is a vastly improved manuscript in content, and certainly demonstrates a substantial body of research.

As mentioned in the first round of review, the quality of English is commendable, but a thorough editing for English would provide noticeable improvement. In general, minor syntactical errors remain, and I make mention of a non-exhaustive list here.

- Example Line 249-250: Before the name of a compound (i.e., stavudine), an article is not appropriate; therefore, mentions of “the stavudine (for example)” should be corrected to “stavudine.” This is only one of a few corrections required.
- Line 251: This is also a general rule for numbered compounds too. “The precursor 4m” should be corrected to “precursor 4m.” This mistake persists through much of the later pages of the manuscript.
- Line 14: “...clinically used...” should be “...clinically-used...”
- Line 16: “Currently, the most widely used methods for synthesizing β - and α -nucleosides involve N-glycosylation or pentose aminooxazoline strategy.” This is awkward wording and should be revised.
- Line 175: “on the gram scale” should be “at gram scale.”
- Line 193: “...was reduced to the alcohol when the NaBH₄-mediated reduction...”
- Line 195: “Added” is probably more suitable than “subjected”; “NaOH aqueous” should be “aqueous sodium hydroxide.”
- Line 196: “solution” should be removed. “Acetonitrile solution” would imply there is something already added to liquid acetonitrile, which is not the case.
- Line 202: “...for two steps” should be “...over two steps.”

- Line 204: “tricyclic key intermediate” should be “key tricyclic intermediate.”
- Line 240: “...through 7 steps” should be “in 7 steps.”
- Line 244: “...under the conditions of...” could be replaced with “...using...”
- Line 245: “deprotection of TBS group...” should be “removal of the TBS group...” The TBS group is not being deprotected, the alcohol is.
- Line 249: Same concept: “removal of the Tr group.”
- Line 254: “one-pot” should be “one pot” in this instance.
- Line 271: “was blocked with a methoxyl group” should be “with a methoxy group.”
- Line 272: “3a were” should be “3a was.”
- Line 287: “without Nal additive” should be “without the Nal additive.”
- Line 288: “Furthermore, we use...” should be “Furthermore, we used.”
- Line 297: “Lewis bases has...” should be “Lewis bases have...”
- Line 307: “Int-IV has a lower energy...” should be “Int-IV has lower energy...”
- Line 321: “...crucial intermediates than...” should be “...crucial intermediates that...”

Responses to the reviewers' comments:

Reviewer 1

The authors have addressed the concerns of the reviewers. I recommend acceptance of the manuscript for publication.

Response: *We thank the reviewer for this positive comment.*

Reviewer 2

I thank the authors for their corrections and additions; at this time, I recommend the article for publication in Nature Communications. This is a vastly improved manuscript in content, and certainly demonstrates a substantial body of research.

As mentioned in the first round of review, the quality of English is commendable, but a thorough editing for English would provide noticeable improvement. In general, minor syntactical errors remain, and I make mention of a non-exhaustive list here.

- Example Line 249-250: Before the name of a compound (i.e., stavudine), an article is not appropriate; therefore, mentions of “the stavudine (for example)” should be corrected to “stavudine.” This is only one of a few corrections required.

Response: *We thank the review for pointing this out. We have deleted “the” before the name of a compound.*

- Line 251: This is also a general rule for numbered compounds too. “The precursor 4m” should be corrected to “precursor 4m.” This mistake persists through much of the later pages of the manuscript.

Response: *We thank the review for pointing this out. We double-checked the manuscript carefully and deleted “the” before numbered compounds.*

- Line 14: “...clinically used...” should be “...clinically-used...”

Response: *We thank the review for pointing this out. The expression “clinically used” has been revised as “clinically-used”.*

- Line 16: “Currently, the most widely used methods for synthesizing β - and α -nucleosides involve N-glycosylation or pentose aminooxazoline strategy.” This is awkward wording and should be revised.

Response: *We thank the review for pointing this out. We have modified this sentence as “Currently, the most widely used methods for synthesizing β - and α -nucleosides are N-glycosylation and pentose aminooxazoline strategy, respectively”.*

- Line 175: “on the gram scale” should be “at gram scale.”

Response: *We thank the review for pointing this out. The expression “on the gram scale” has been revised as “at gram scale”.*

- Line 193: “...was reduced to the alcohol when the NaBH₄-mediated reduction...”

Response: *We thank the review for pointing this out. This sentence has been modified as “When the NaBH₄-mediated reduction of **5a** was carried in 30 min, only benzyl carboxylic group in **5a** was reduced to yield alcohol intermediate, which was Tr-protected to give **5i** in 84% yield over two steps.”*

- Line 195: “Added” is probably more suitable than “subjected”; “NaOH aqueous” should be “aqueous sodium hydroxide.”

Response: *We thank the review for pointing these out. This sentence has been modified as “Compound **5i** was added into aqueous sodium hydroxide and acetonitrile to give **11**”.*

- Line 196: “solution” should be removed. “Acetonitrile solution” would imply there is something already added to liquid acetonitrile, which is not the case.

Response: *We thank the review for pointing this out. This sentence has been modified as “Compound **5i** was added into aqueous sodium hydroxide and acetonitrile to give **11**”.*

- Line 202: “...for two steps” should be “...over two steps.”

Response: *We thank the review for pointing this out. The expression “for two steps” has been modified as “over two steps”.*

- Line 204: “tricyclic key intermediate” should be “key tricyclic intermediate.”

Response: *We thank the review for pointing this out. The expression “tricyclic key intermediate” has been modified as “key tricyclic intermediate”.*

- Line 240: “...through 7 steps” should be “in 7 steps.”

Response: *We thank the review for pointing this out. The expression “through 7 steps” has been revised as “in 7 steps”.*

- Line 244: “...under the conditions of...” could be replaced with “...using...”

Response: We thank the review for pointing this out.

• Line 245: “deprotection of TBS group...” should be “removal of the TBS group...” The TBS group is not being deprotected, the alcohol is.

Response: We thank the review for pointing this out. The expression “deprotection of TBS group” has been modified as “removal of the TBS group”.

• Line 249: Same concept: “removal of the Tr group.”

Response: We thank the review for pointing this out. The sentence has been changed to “After removal of the Tr group, stavudine was obtained”.

• Line 254: “one-pot” should be “one pot” in this instance.

Response: We thank the review for pointing this out. The expression “one-pot” has been revised as “one pot”.

• Line 271: “was blocked with a methoxyl group” should be “with a methoxy group.”

Response: We thank the review for pointing this out. The sentence has been changed to “To be specific, when the acidic site (-OH) of **CI** was with a methoxy group, (Supplementary Table 7, entries 6 and 20), the corresponding product **3a** was nearly racemic (<5% ee)”.

• Line 272: “3a were” should be “3a was.”

Response: We thank the review for pointing this out. The expression “**3a** were” has been revised as “**3a** was”.

• Line 287: “without NaI additive” should be “without the NaI additive.”

Response: We thank the review for pointing this out. The expression “without NaI additive” has been revised as “without the NaI additive”.

• Line 288: “Furthermore, we use...” should be “Furthermore, we used.”

Response: We thank the review for pointing this out. The sentence has been changed to “Furthermore, we used interaction region indicator (IRI)^{56,57} and fuzzy bond order (FBO)^{58,59} to analyze the interactions between atoms of Int-I (IRI pic. of Int-I in Figure 8a)”.

• Line 297: “Lewis bases has...” should be “Lewis bases have...”

Response: We thank the review for pointing this out. The expression “Lewis bases has” has been revised as “Lewis bases have”.

• Line 307: “Int-IV has a lower energy...” should be “Int-IV has lower energy...”

Response: We thank the review for pointing this out. The expression “Int-IV has a lower energy” has been revised as “Int-IV has lower energy”.

• Line 321: “...crucial intermediates than...” should be “...crucial intermediates that...”

Response: We thank the review for pointing this out. The sentence has been changed to “Meanwhile, pyrimidine nucleoside products bearing iodine at their C2’ positions were shown to be crucial intermediates that can be further functionalized to yield structurally diverse nucleoside analogs”.